# Exorbital Lacrimal Gland Ablation and Regrafting Induce Inflammation but Not Regeneration or Dry Eye

**DOI:** 10.3390/ijms25158318

**Published:** 2024-07-30

**Authors:** Adriana de Andrade Batista Murashima, Ariane M. S. Sant’Ana, Jacqueline Ferreira Faustino-Barros, Elísio B. Machado Filho, Lilian Costa Mendes da Silva, Marina Zilio Fantucci, Carolina Maria Módulo, Fernando Chahud, Denny Marcos Garcia, Eduardo M. Rocha

**Affiliations:** 1Department of Ophthalmology, Otorhinolaryngology and Head & Neck Surgery, Ribeirao Preto Medical School, University of Sao Paulo, Ribeirao Preto 14049-900, SP, Brazil; 2Department of Pathology and Legal Medicine, Ribeirao Preto Medical School, University of Sao Paulo, Ribeirao Preto 14049-900, SP, Brazil

**Keywords:** inflammation, lacrimal functional unit, regeneration

## Abstract

The study evaluated the regenerative responses of the lacrimal functional unit (LFU) after lacrimal gland (LG) ablation. The LG of Wistar rats was submitted to G1) partial LG ablation, G2) partial ablation and transplantation of an allogeneic LG, or G3) total LG ablation, (n = 7–10/group). The eye wipe test, slit lamp image, tear flow, and histology were evaluated. RT-PCR analyzed inflammatory and proliferation mediators. The findings were compared to naïve controls after 1 and 2 months (M1 and M2). G3 presented increased corneal sensitivity, and the 3 groups showed corneal neovascularization. Histology revealed changes in the LG and corneal inflammation. In the LG, there was an increase in MMP-9 mRNA of G1 and G2 at M1 and M2, in RUNX-1 at M1 and M2 in G1, in RUNX-3 mRNA at M1 in G1, and at M2 in G2. TNF-α mRNA rose in the corneas of G1 and G2 at M2. There was an increase in the IL-1β mRNA in the trigeminal ganglion of G1 at M1. Without changes in tear flow or evidence of LG regeneration, LG ablation and grafting are unreliable models for dry eye or LG repair in rats. The surgical manipulation extended inflammation to the LFU.

## 1. Introduction

It is controversial whether the main lacrimal gland (LG) is an indispensable organ for vision and ocular surface integrity [1,2,3]. However, it is clear that the lacrimal functional unit (LFU) responds together to challenges to one or more of its components and that evolutionally closer animals present similar responses to those challenges [2,3,4,5,6].

Damage to the main LG in humans and primates impacts other tissues of the LFU and induces dry eye disease (DED). However, it seems to be attenuated in the long term [1,7]. On the other hand, exorbital LG ablation in rodents has been used as animal models in short-term studies to evaluate the efficacy of novel topical eye drop therapies for DED; however, the capacity for rodent exorbital LG spontaneous auto regeneration is unknown [8,9].

The potential of the mammalian LG for auto or therapeutic-driven repair is the subject of many doubts and may be influenced by different damage mechanisms and the time-length of observations [10,11]. Lacrimal gland damage induced by prolonged inflammation in the course of Sjögren’s Disease, cancer, radiotherapy, surgical, or traumatic damage to the innervation or the LG itself, among other conditions, induces permanent functional impairment atrophy and fibrosis, despite the presence of regenerative conditions [10,11,12]. The limitations to addressing why LG persists with a post-mitotic and nonregenerative status, even presenting the machinery for regeneration, justify the present work.

Lacrimal gland dysfunction can arise from different diseases or local injuries [4,12]. However, tissue ablation via surgery or radiotherapy may also elicit declines in LG support of the lacrimal functional unit (LFU), which deserves a more profound understanding [5,13,14,15].

There is evidence that autologous salivary gland or neonatal LG transplantation can restore sufficient LG structure and function to be therapeutic in dry eye models or patients [16,17,18,19]. However, it is unknown whether or not increases in LG secretion can be attributed to proliferative or repair events, or if it is just fluid exudation related to inflammatory healing as a response to the surgical procedure, and whether this secretory activity persists for extended periods [8,20,21,22].

The LG’s impact on the LFU and its repair capacity is species-specific and modulated by pro-mitotic factors [6,10]. Removal, recession, extraction, excision, and ablation are similar terms used in medical science literature when referring to surgical destruction of an organ. Here, we adopted the word ablation to describe the proposed partial or complete manipulation of the LG.

In humans, main LG agenesia, atrophy, or ablation induces persistent dry eye syndrome (DES) [1,23,24,25,26]. In monkeys, the complete ablation of the main LG led to a reduction in tear flow and ocular surface staining. The findings gradually reverted over 20 weeks, probably supported by the accessory glands [13]. In rats, LG total ablation is used to establish a DED model to search for novel therapeutic options, pharmacological and surgical, based on the belief that it is an irreversible and reproducible model [8,27]. Recently, the exorbital LG total ablation mouse model was more aggressive to the ocular surface than drug-induced cholinergic blockage combined with a dry environment [28]. Rabbit models are not investigated for the LG regeneration model as much as rats and mice; however, the larger organs offer an exciting model for surgical and grafting studies, where the evaporative and tear deficiency repercussions can be accessed [29].

In the mouse LG, injury-induced tissue inflammation is associated with apoptosis and autophagy [30]. Moreover, it changes protein expression and growth factors relevant to LG morphogenesis and regeneration. They include fibroblast growth factor-10 (Fgf10), bone morphogenetic protein 7 (Bmp7), and signaling proteins, such as nestin and Pax6 [30,31,32]. Their presence further substantiates the notion that following the damage, the regeneration would occur in three phases (i.e., inflammation, new tissue formation, and remodeling), where LG cells require molecular machinery and local stem cells to undergo proliferation and differentiation [19,33]. A recent review addressed the topic of LG regeneration, which involved several mediators, including Bmp-7, and stem cells, mentioning various species but not in rat models [33].

The study investigates LG ablation. The conflicting information mentioned above reveals that LG ablation or direct damage needs to be understood more regarding DED aspects, natural regeneration, and extension of the functional damage. Thus, we submitted rats to three different models of LG ablation to investigate the detailed structural and functional repercussions on the LG and the LFU. We also searched for a specific potential pathway relevant to the disease mechanism. This information could be helpful in a subsequent treatment approach, such as an adult LG or organoid transplant, LG gene therapy, or CRISPR to promote genetic edition and pro-mitotic agent reprogramming to restore the LG and the LFU homeostasis [19,30,31,32,33].

Our hypothesis predicts that LG ablation is irreversible and leads to structural, molecular, and functional repercussions in the LFU. Still, partial LG ablation and grafting will reveal the proliferative agents and structural attempts to regenerate the LG.

We describe here different rat models of LG damage obtained via partial ablation (G1), allogeneic LG transplantation following partial ablation (G2), complete ablation (G3), and a control, where the changes induced in LG structure and function and LFU health were compared. Furthermore, the possible signs of repair were evaluated in response to those interventions.

## 2. Results

The three distinct models of LG damage are shown and compared with naïve controls to allow inference with clinical conditions that simulate partial ablation (G1), partial damage with attempted repair with allogeneic LG transplant (G2), or total LG ablation (G3).

### 2.1. Lacrimal Gland Partial Ablation (G1)

#### 2.1.1. In Vivo Exams

After 1 and 2 months, the corneal sensitivity did not change compared to the control group’s (Table 1). The slit-lamp exam revealed corneal neovascularization and punctate keratitis (Figure 1, picture 1 upper lane and picture 1’ lower lane).

The results of a phenol red thread test mean wetting in mm indicates that those tear secretion measurements of G1 were similar to control values over the observational period (Table 1).

Corneal epithelial cell impression cytology did not reveal metaplasia compared to controls at 1 or 2 months (*p* > 0.05, Fisher test).

As expected, LG weight in G1 was significantly lower than controls (Table 1). The results indicate that LG weight after 1 or 2 months remained lower and did not recover the weight of the intact LG.

#### 2.1.2. Ex Vivo Assays

Proliferating cell nuclear antigen (PCNA) expression levels were measured to determine whether or not any of the invasive procedures induced changes in LG proliferation. Such levels remained similar in the LG of G1, G2, and control groups throughout the two months (n = 5 rats/group per time point).

The LG histology comparing control and G1 revealed that LG acinar and ductal structures were unchanged, except for mild cytoplasmic vacuolation and atypical nuclei suggestive of attempted mitosis or apoptosis after months 1 and 2 (Figure 2 and Figure 3). Corneal, conjunctival, and lid histology was not changed by partial LG ablation in G1 (Figure 4 for corneal histology in month 2).

The RT-PCR analysis of Mmp-9 mRNA levels in G1 rose at months 1 and 2 (*p* = 0.0298 and 0.0107, respectively) (Figure 5B). Mmp-9 mRNA levels were raised at months 1 and 2 in G2 (*p* = 0.0094 and 0.0107, respectively) (Figure 5B). The mRNA expression of IL-1β and IL-6 did not significantly change in the LG of any groups and time points compared to the control group.

The RT-PCR of the mRNA expression of proliferative and tissue repair mediators in LG of G1 revealed that SMAD1 mRNA levels rose in month 2 (*p* = 0.0472) (Figure 5C), RUNX-1 mRNA levels rose in G1 in month 2 (*p* = 0.0138) and RUNX-3 mRNA levels rose in LG of G1 at month 1 and 2 (*p* = 0.0182 and 0.0298, respectively) and in G2 at month 1 and 2, compared to controls (*p* = 0.042 and 0.0182, respectively) (Figure 5D,E). The mRNA of FgF10 and Bmp7 did not change in the LG of any groups and time points compared to the control group.

The RT-PCR assay to compare the expression of inflammatory cytokine mRNA in the cornea and investigate the potential inflammatory repercussions of those interventions on the ocular surface, as indicated via the slit lamp and histological analysis, revealed that G1 presented higher levels of TNF-α mRNA at month 1 (*p* = 0.0298) (Figure 6A). The mRNA expression of IL-1β, IL-6, and MMP9 did not significantly change in any groups and time points compared to the control group.

In addition, considering that LG intervention increased the corneal sensitivity in G3, we investigated the changes in pro-inflammatory cytokine mRNA expression in trigeminal ganglions (TGs) compared to the control group at 1 and 2 months after the intervention. The mRNA expression of IL-1β rose in the TG of G1 at month 2 (*p* = 0.0094) (Figure 6B) and MMP9 decreased in the TG of G3 at month 2 (*p* = 0.0497) (Figure 6C). The other cytokine mRNA tested (IL-6 and TNF-α) presented no changes in TG compared to the control group at 1 or 2 months.

### 2.2. Lacrimal Gland Partial Ablation and Allogeneic Transplant (G2)

#### 2.2.1. In Vivo Exams

Corneal sensitivity challenged by capsaicin and measured via the eye wipe test revealed a similar number of forepaw eye wipes comparing G2 and controls (Table 1). However, partial LG ablation followed by grafting showed pronounced corneal neovascularization in the slit lamp exam at month 1 (Figure 1, upper lane). Furthermore, tear secretion measurements (Table 2) and corneal epithelial cell impression cytology results were similar to controls at 1 or 2 months (*p* > 0.05, Fisher test).

The LG weight also presented significantly lower values than controls (Table 1), showing no evidence of tissue mass recovery to the intact LG levels (Table 1).

#### 2.2.2. Ex Vivo Assays

In G2, PCNA levels remained similar to those of the control group LG throughout the two months.

Histological evaluation of the G2 H&E-stained LG at month 1 revealed that the transplanted tissue retained some of its attributes. However, the ductal structure dilated, leukocyte infiltration occurred, and the acinar structure was lost in the grafted area. The host LG of G2 was similar to the control (Figure 2).

At month 2, the histology changes in G2 LG became more pronounced. The ducts were more dilated, acinar cells were absent, the parenchyma was replaced by fibrosis, and there was leukocyte infiltration. The vascular and ducts were preserved only in the interlobular bundle (Figure 3).

Despite no striking corneal histology changes at month 1, inflammatory cell infiltration and micro-vessels appeared in the corneal stroma at month 2 (Figure 4). These observations agree with the results of the in vivo slit lamp exam of the rat corneas at month 1 (Figure 1). Also, at month 2, conjunctival and lid histology were well preserved, as indicated by intact epithelial and associated structures.

The RT PCR analysis of the mRNA expression of inflammatory mediators in the LG revealed higher levels of TNF-α mRNA in G2 at month 2 (*p* = 0.0298) (Figure 5A). Levels of MMP-9 mRNA were raised at months 1 and 2 in G2 (*p* = 0.0094 and 0.0107, respectively) (Figure 5B).

The RT-PCR of the mRNA expression of proliferative and tissue repair mediators in the LG of G2 at months 1 and 2 was compared to controls (*p* = 0.042 and 0.0182, respectively) (Figure 5E).

The RT-PCR assay of inflammatory cytokine mRNA in the cornea revealed G2 higher levels of TNF-α mRNA at month 2 (*p* = 0.0094) (Figure 6A) in agreement with the inflammatory repercussions of those interventions on the ocular surface, as indicated via the slit lamp and histological analysis.

No changes in pro-inflammatory cytokine mRNA expression in TG compared to the control group at 1 and 2 months after the intervention were observed in G2 (Figure 6B,C).

### 2.3. Lacrimal Gland Total Ablation (G3)

#### 2.3.1. In Vivo Exams

The eye wipe test, which occurred in the sequence of capsaicin eye drops, showed a higher mean number of forepaw eye wipes in the G3 at month 2 (*p* = 0.0438) (Table 1), indicating increased corneal sensitivity.

Slit lamp examination revealed no changes in the G3 cornea appearance (Figure 1, upper lane), despite presenting mild punctate keratitis under a cobalt filter exam with 2% sodium fluorescein dye staining (Figure 1, lower lane). Those observations were stable at month 2 but included mild peripheral corneal neovascularization.

The phenol red thread test mean wetting showed similar tear flow to controls (Table 1). Again, corneal epithelial cell impression cytology did not reveal any metaplasia among the groups compared to controls at months 1 or 2 (*p* > 0.05, Fisher test).

The LG weight was not registered since it was removed at the beginning of the experimental period, and the surgical site was empty at the end of both months.

#### 2.3.2. Ex Vivo Assays

Since the whole exorbital LG was ablated at the beginning of the experimental period, PCNA expression levels in LG and LG histology after the experimental period of 1 and 2 months were not investigated in G3. Moreover, no striking corneal histology changes were observed in G3 at month 2 (Figure 4) or in the conjunctival and lid histology at month 2.

The RT-PCR analysis of the mRNA expression of inflammatory or proliferative mediators in the LG of G3 was not investigated in G3 once more because the total ablation of the LG did not preserve the minimum in vivo remaining sample for analysis.

The RT-PCR assay to compare the expression of inflammatory cytokine mRNA in the cornea confirmed the lack of inflammatory repercussions of total LG ablation on the ocular surface, as indicated by the similar levels of mRNA expression of TNF-α (Figure 6A) and MMP-9, IL-1β, and IL-6 compared to the control group corneas.

Considering that LG total ablation increased the corneal sensitivity in G3, we investigated the changes in pro-inflammatory cytokine mRNA expression in TG compared to the control group at 1 and 2 months after the intervention. The only change was that MMP-9 mRNA expression was reduced at month 2 (*p* = 0.0497) (Figure 6C). The other cytokine mRNA tested (IL-1β, IL-6, and TNF-α) presented no changes in TG compared to the control group at 1 or 2 months.

## 3. Discussion

This study was planned to answer two major questions. The first one is whether exorbital LG ablation is a reliable animal model for dry eye. Second, whether spontaneous or graft-induced regeneration occurs after three different kinds of planned surgical manipulation. Moreover, we investigated those procedures’ clinical, histological, and molecular journeys.

The most relevant findings are the changes in corneal sensitivity and neovascularization in response to the surgical manipulations of the LG. Inflammatory mediator mRNA was raised in the LG, the cornea, and the TG. Those findings demonstrated the extra glandular extension of those interventions. They follow previous descriptions of regional and systemic repercussions of LG losses in mice. A recent study observed the response to LG complete ablation in mice and reported changes on the contralateral side, including inflammation of the LG, keratitis, and a reduction in conjunctival goblet cells [34]. Another study with LG complete ablation in mice induced an increase of IL-1β, TNF-α, and MMP-9 mRNA in the ipsilateral cornea [28]. Mouse behavior changed after LG’s complete ablation, including a reduction in libido and anxiety and increased corneal sensitivity, and those changes were different when comparing male and female sex [35].

Differently from data obtained from mice, the rat exorbital LG ablation induced only mild keratitis and unchanged tear flow at 1 and 2 months. These observations do not agree with previous descriptions of tear flow reduction in 50% in male Sprague–Dawley (SD) rats 8 weeks after exorbital LG ablation and caution against the use of the rat model of LG ablation in dry eye comparative therapeutic studies [27]. The reduction in tear flow is superior to 30% in rats after LG complete ablation was described when both the exorbital and intraorbital LGs are removed [36].

The regenerative capacity of the LG was not observed here, although it is conceivable since rodent LG cells grow in culture and present markers of identifiable local stem cells [33,37,38]. In our study, complete or partial ablation did not respond to tissue regeneration, PCNA levels in the LG were similar to controls, and tissue weight did not increase over two months. Furthermore, the study showed that allogeneic LG grafting is not tenable as a therapeutic option for rat LG repair since this procedure was not tolerated based on increases in structural change and inflammation in the transplanted tissue, which is different in the method and response to submandibular salivary gland grafting in SD male rats. The exorbital LG removal created a dry eye model where the graft integrated and maintained its structure and function [8]. The overexpression of RUNX 1 and 3 and MMP9 mRNA in the partially ablated LG (G1) suggested an attempt of tissue repair. Still, the lack of response of other mediators, such as BMP7 and FGF, also involved in LG organogenesis and regeneration, suggests that inductive actions are necessary [33,39,40]. This adverse finding indicates the need to develop novel procedures to overcome LG graft failure, probably involving LG organoid transplantation.

In mice, IL-1α stimulates LG inflammation but also allows its repair; fewer therapeutic effects were observed after LG duct ligation [30,41,42]. It may indicate a selective response to different methods of damage, distinct animal species response, and the time length among the variables involved in the delicate process of LG regeneration. 

Overall, the present work reveals that LG damage that simulates trauma, surgical intervention, radiotherapy, or long-term disease, as observed in SjD or GvHD that terminate with LG atrophy or fibrosis, is dominated by prolonged inflammatory mediators, such as TNF-α and MMP-9 and others, and extension of the inflammation to the LFU. Conversely, the expression of pro-mitotic agents in the LG is modest or null, not allowing the regenerative process to progress. The inductive intervention capable of reverting the imbalance between inflammation and regenerative events, skilled to promote LG regeneration, so far involves in vitro LG organogenesis, using different lineages of allogeneic embryonic stem cells stimulated by humoral factors and further grafting in the location of the damaged LG [19,39]. This strategy to promote pro-mitotic factors and downregulate inflammatory mediators has been proven to be plausible in rodents, although very complex. In the future, strategies involving local regularly interspaced cluster shortly palindromic repeats gene edition using Cas-9 enzyme (CRISPR/Cas9) could support promising therapies to improve the regenerative process and include the expression of Pax6 to allow differentiation in adult LG. However, before considering them for human therapies, CRISPR/Cas9 still needs to be tested in animal models [43].

In studies for DED therapy, the chosen animal model must mimic the clinical signs associated with a DED of interest. In this regard, models induced by hormone deprivation and neural damage, sensory or autonomic, are helpful in mimicking clinical conditions, the mechanisms, and proposed therapeutic strategies [4,44]. The present work reveals that exorbital LG ablation does not induce DED, probably because the other LGs can sustain the demand for tear secretion. 

In the future, if LG regeneration is promoted in response to severe tissue damage, whether induced by trauma, surgical removal, or radiotherapy, among other conditions, the most promising strategies will include LG organogenesis and bioengineering [19,39]. Before this can be undertaken in a broad sense, a better understanding is needed of the interactions among morphogenesis-related proteins and their complex protective and disruptive balance in the physiopathology of exocrine diseases as recently demonstrated by the role of BMP6 in the Sjögren’s syndrome [45].

The limitations of the present work are the absence of more time-points of evaluation, including the short-term outcome, and the absence of a comparative analysis of rats of the female sex under those conditions. Another limitation is associated with the rudimentary technique applied to test LG graft integration and its role as an inductor of regeneration. A simple appose of part of the LG obtained from a sibling and closing the skin wound with cyanoacrylate glue does not meet the LG healing in the present work as previously described [14]. Bioengineered LG grown in vitro and customized and placed in the diseased eye appears to be the clue for each causative severe DED condition [46].

In conclusion, exorbital LG ablation did not induce DED or LG repair in rats in this study’s ablation models. The possible reasons are the inflammatory disruption caused by the LG damage and the destabilization of the LFU. It may induce local conditions that inhibit the regenerative process.

## 4. Materials and Methods

### 4.1. Experimental Design and Animal Procedures

All experimental procedures adhered to the ARVO Statement for the Use of Animals in Ophthalmic and Vision Research and were approved by the committee on animal experimentation of the School of Medicine at Ribeirão Preto, University of São Paulo (number 109/2008).

Eight-week-old male Wistar rats (Rattus norvegicus) were obtained from the Animal Breeding Centre of the School of Medicine at Ribeirão Preto, University of São Paulo (FMRP-USP), Ribeirão Preto, SP, Brazil. Animals were given free access to standard rodent chow and water and randomly separated into 4 study groups.

The surgical interventions and the in vivo comparative studies were performed after intraperitoneal anesthesia with a combination of ketamine (5 mg/100 g b.w.) (União Química Farmacêutica S.A, Embu-Guaçu, SP, Brazil) and xylazine (2 mg/100 g b.w.) (Laboratorio Callier S.A., Barcelona, Spain), after ensuring the absence of corneal and caudal reflexes.

Exam data and samples were compared among the four following experimental rat groups:

Naïve Control (C): the eyes and extraorbital lacrimal glands were kept intact. In vivo exam data were compared with the other groups, and the eye globe and extraorbital LG were used as a control in laboratory assays (n = 7 rats).

Group 1 (G1): aseptic surgical removal of anterior half part of the right extraorbital LG, with a cut (5 mm in length) through the skin in a lateral area of the head 3 mm equidistant between the eye and ear, followed by two cyanoacrylate glue drops (Locite, Henkel Ltda, Diadema, SP, Brazil) to close the skin margins, and covered with a single, 5 mm application of antibiotic and anti-inflammatory ointment (Cylocort, União Química Farmacêutica Nacional S.A, Brasilia, DF, Brazil). This animal model intervention was included to investigate the regenerative capacity and the functional and molecular responses associated with the intervention (n = 10 rats).

Group 2 (G2): the aseptic surgical removal procedure was the same as for group 1. Subsequently, an allogeneic transplant from a twin brother was performed on LG tissue, which had approximately the same size, and it was embedded in the extirpated site. The LG was removed from the anesthetized donor; then, its capsule was opened and excised; the tissue was cut with a surgical blade in the Petry dish and washed with saline. The LG transplantation was completed when the LG graft was juxtaposed in the same topographical site of the receptor animal, also anesthetized, after a similar piece of LG tissue was removed. After bleeding control, cyanoacrylate glue was applied to the borders of the skin to close the skin wound, followed by a 5 mm application of the same antibiotic and anti-inflammatory ointment (n = 8 rats). The rationale was to investigate the possibility of the grafted tissue regarding histological and functional restoration.

Group 3 (G3): aseptic skin incision and exposure of right extra orbital LG, followed by complete LG ablation and homeostasis of the vascular beds. Afterward, the skin was glued and treated as mentioned above (n = 11 rats). The skin wounds were undetectable two days later in all animals of the three groups. The rationale was to investigate the impact of the absence of the LG on the ocular surface and the possibility of the local germination of a novel LG and its functional restoration.

Animals were evaluated after 1 and 2 months of the procedure (n = 4–7 rats/group per time point).

#### 4.1.1. In Vivo Exam

At months 1 and 2 after surgical intervention, the body weights of the animals were registered.

#### 4.1.2. Eye Wipe Test

At the end of the two experimental periods (1 m and 2 m) for each group (i.e., C, G1, G2, and G3), the rats were submitted to the eye wipe test in response to capsaicin (CAP) to investigate the corneal sensitivity. Following acclimation of the animals to Plexiglas transparent chambers for one hour, the right eyes of all models were treated with the instillation of 20 μL of 10 μM CAP diluted in PBS at pH 7.2 and 25 °C (Sigma-Aldrich Brasil Ltda., Cotia, SP, Brazil).

The eye wipe behavior was recorded with a digital camera (DSC-W5, Sony, Japan) for 5 min after the instillation of CAP. Eye wipe movements with the forepaws (EWT) registered along 3 min, starting 1 min after the CAP eye drop, were counted afterward from the digital movies recorded for each rat by a masked observer using an iMac computer (Apple Inc, Cupertino, CA, USA). The means were compared among the groups.

Afterward, the rats were anesthetized with the same drugs and doses used for surgical interventions, and the eyes were examined with a slit lamp (Carl Zeiss, Jena, Germany). The corneal epithelial integrity was evaluated on a slit lamp and cobalt filter after 2% sodium fluorescein dye staining. The punctate keratitis was graded from 0 to 15, as previously described [47].

Tear flow was measured in millimeters with the phenol red thread (PRT) for 30 s, and the values obtained were compared among the groups (Showa Yakuhin Kako Co; Ltd., Tokyo, Japan & Menicon USA Inc., Clovis, CA, USA).

#### 4.1.3. Tissue Harvesting and Storage for Analysis

Extraorbital LG, eye globes with lids, and trigeminal ganglions (TGs) were removed at the end of months 1 and 2 of the experimental period. The tissues were weighed, and the samples were processed for histology, qRT-PCR, and Western blotting, as detailed below. The rats were euthanized with excess anesthesia (ketamine 15 mg/100 g of body weight and xylazine 6 mg/100 g of body weight) and sodium thiopental (1000 mg/kg) (Laboratório Cristália, São Paulo, SP, Brazil).

#### 4.1.4. Impression Cytology

Cornea epithelial cells were harvested from the temporal area with 0.45 µm pore size filter paper for impression cytology (Millipore, Billerica, MA, USA), at months 1 and 2.

Corneal epithelial cells adherent to the filter paper were fixed with a 70% ethanol, glacial acetic acid, and formalin solution and stained with periodic acid–Schiff (PAS) and hematoxylin. They were transferred to microscope slides. Squamous metaplasia of epithelial cells was staged in a masked manner according to a four-stage classification from 0 (normal morphology) to 3 (squamous metaplasia) based on the coloration and cellular appearance: nuclei size and presence of mucous secretion. The images were analyzed with a light microscope (Olympus BX40) and photographed with a digital camera (Olympus Q-color 5) at 100 and 400× magnification (Olympus Corporation, Tokyo, Japan).

#### 4.1.5. Western Blotting

The LG was homogenized with a polytron (Virsonic, Biopharma, Winchester, UK) in a buffer containing 50 mM Tris, pH 7.5, 500 mM NaCl, 0.1% Triton, and protease inhibitor cocktail set III (Calbiochem, San Diego, CA, USA). The PCNA expression levels in whole cell lysates were evaluated to assess cell proliferative activity.

Protein from homogenized tissue was measured via the biuret µdye test. Samples were treated with Laemmli buffer, and equal amounts of protein per sample (70 µg) were subjected to SDS-PAGE (10% Tris-acrylamide) in a Bio-Rad miniature lab gel apparatus (Miniprotean, Bio-Rad Laboratories, Richmond, CA, USA), in parallel with pre-stained protein standards and dithiothreitol (Bio-Rad, Hercules, CA, USA). Proteins were then electro-transferred from the gel to a 0.22 µm nitrocellulose membrane (Bio-Rad, Hercules, CA, USA) for 2 h at 120 V in a Bio-Rad miniature transfer apparatus (Miniprotean). After blocking, the membranes were incubated overnight in a buffer containing 3% bovine serum albumin (BSA) using anti-PCNA or alpha-tubulin antibodies (Table 2) and then washed three times with tris buffered saline/tween (TBS/T). Detection was performed using enhanced chemiluminescence (SuperSignal West Pico, Pierce, Rockford, IL, USA) after a 2 h incubation with a horseradish peroxidase-conjugated secondary antibody (1:10,000, Invitrogen, São Paulo, SP, Brazil). Band intensities were quantified via optical densitometry (Scion Image, Frederick, MD, USA), using α-tubulin as an internal control.

#### 4.1.6. Histology

Cornea and LG samples for histology were collected and frozen in OCT compound (Sakura FineTek Inc., Torrance, CA, USA). After tissue cutting (6 μm) and transference to slides, the 10th to the 14th corneal and LG sections were submitted to hematoxylin/eosin staining (5 samples per animal, n = 5/group, per time point). Digital photos were obtained (Nikon Eclipse E800, Nikon USA, Melville, NY, USA) under a microscope (microscope specification), and the structure and area of the LG were compared in a masked manner.

#### 4.1.7. Quantitative Real-Time PCR

Cornea, LG, and TG tissues from the right side of the rats of the four groups were harvested, embedded in RNA stabilization solution (RNAlater Solution, Ambion, Waltham, MA, USA), and stored at −80 °C until proceeding to RNA extraction, quantification, quality evaluation, and real-time quantitative PCR (qPCR) analysis. The relative expression of the mRNA of proinflammatory cytokines Il1b, Il6, Tnf, and Mmp9 was compared in LG, CO, and TG samples. In addition, in LG, the relative expression of the mRNA of the tissue repair elements Bmp7, Runx1, Runx3, Fgf10, and Smad1 was compared using commercial primers and beta-actin mRNA as an internal control (Life Technologies, Carlsbad, CA, USA).

Total RNA samples were extracted from the tissues using PureLink MiniKit (Ambion by Life Technologies, USA) and the kit DNA-Free DNase Treatment & Removal (Ambion by Life Technologies, USA) to remove genomic DNA contamination, according to the manufacturer’s instructions. RNA was quantified with a spectrophotometer NanoDrop 2000 c (Thermo Scientific, Wilmington, DE, USA).

Samples containing 500 ng of CO total RNA, 1000 ng of LG total RNA, and 350 ng of TG total RNA were used to synthesize the cDNA with the QuantiTect Reverse Transcription Kit (Qiagen, Germantown, MD, USA) in the ProFlex PCR System (Applied Biosystems, Carlsbad, CA, USA).

The qPCR was performed using a ViiA7 Real-time PCR System (Applied Biosystems, Carlsbad, CA, USA). The following hydrolysis probes were used in this study: Rn.PT 5838028824 (Il1b), Rn.PT 5813840513 (Il6), Rn.PT 5811142874 (Tnf), Rn.PT 587383134 (Mmp9), Rn.PT 5810180444 (Bmp7), Rn.PT 5810814634 (Fgf10), Rn.PT 589220704.g (β-actin) (all these from IDT); Rn00565555_m1 (Smad1), Rn00569082_m1 (Runx1), Rn00590466_m1 (Runx3) (Applied Biosystems, Carlsbad, CA, USA). Each amplification reaction was duplicated with 5.5 uL of the QuantiNova Probe PCR Kit (Qiagen, Germantown, MD, USA), 0.5 μL of hydrolysis probe, and 4.5 μL of a 1:4 dilution of the cDNA in a total volume of 10 μL. The cycles for real-time PCR were as follows: one cycle of 95 °C for 2 min, 50 cycles of 5 s at 95 °C, and 19 s at 60 °C.

The relative quantification was determined using Thermo Fisher Cloud Software, RQ version 3.7 (Life Technologies Corporation, Carlsbad, CA, USA).

### 4.2. Statistical Analysis

Data are reported as the mean ± SEM. The normal distribution of the continuous data was investigated using the Shapiro–Wilk test. Since the data deviated from a normal distribution, the comparisons between groups 1–3 and the control and between the baseline versus 1 and 2 months of observation, were made using the Mann–Whitney U test for continuous data. The Fisher exact test was applied for categorical data, comparing each group with the corresponding control group (GraphPad 5.0 software; Prism, San Diego, CA, USA). Densitometry values of Western blotting were normalized to alpha-tubulin and reported as a ratio to validate loading equivalence. The RT-PCR values of each probe were compared as a ratio of β-actin for each tissue, condition, and time-point. The level of significance was set at *p* < 0.05.

## 5. Conclusions

In conclusion, exorbital LG ablation did not induce DED or LG repair in rats in this study’s ablation models. The present work has limitations regarding the animal model, age and sex chosen, and follow-up length, which must be considered in future studies. Nevertheless, the differences in inflammatory responses and tissue injury are damage-model-type-dependent, not limited to the LG but extended to the TG and cornea. Such variability may have subtle effects on ocular surface health. It should support decisions on future methodological decisions for studies on the topic of the dry eye model and LG regeneration. The future human adult LG regeneration approach should consider multiple strategies, including control of the inflammation and their consequences to the LFU and organogenesis with gene editing.

## Figures and Tables

**Figure 1 ijms-25-08318-f001:**
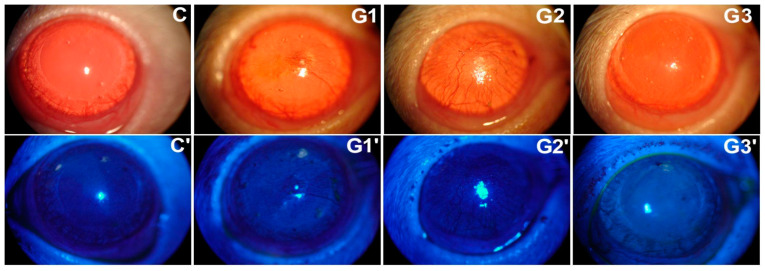
(**Upper lane**); Slit lamp observation of the ocular surface of rats, submitted or not to LG intervention. The photos illustrate the outcome at a 1-month follow-up of eyes from controls (**C**), LG partial ablation (**G1**), ablation followed by allogeneic LG transplant (**G2**), and total LG ablation (**G3**). (**Lower lane**): Slit lamp observation of the ocular surface of rats, submitted or not to LG intervention, using a cobalt light filter and fluorescein dye. The photos illustrate the outcome at 1-month follow-up of eyes from controls (**C’**), LG partial ablation (**G1’**), ablation followed by allogeneic LG transplant (**G2’**), and total LG ablation (**G3’**).

**Figure 2 ijms-25-08318-f002:**
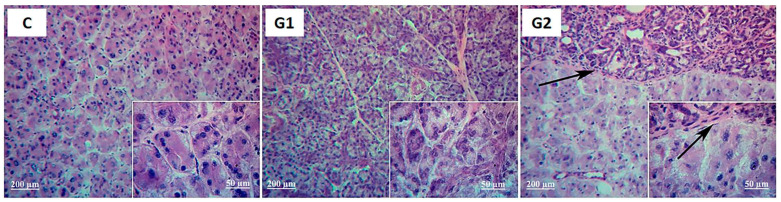
Representative histological images of LG of controls (**C**), LG partial ablation (**G1**), and ablation followed by allogeneic LG transplant (**G2**) after 1 month. Samples were stained with H&E. Magnification, 100× for the figures and 400× for inserts that show details of the acinar cells structure (**right lower corner**). Black arrows indicate the border between the graft (**upper side**) and host (**lower side**) LG.

**Figure 3 ijms-25-08318-f003:**
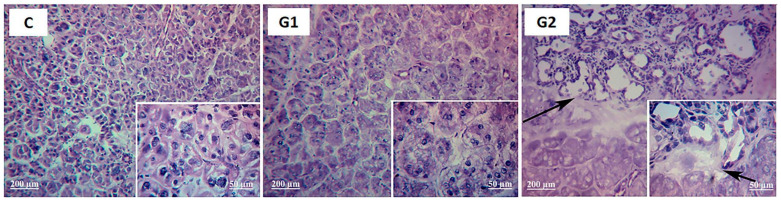
Representative histological images of LG of controls (**C**), LG partial ablation (**G1**), and ablation followed by allogeneic LG transplant (**G2**) after 2 months. Samples were stained with H&E. Magnification, 100× for the figures and 400× for inserts that show details of the acinar cells structure (**right lower corner**). Black arrows indicate the border between graft (**upper side**) and host (**lower side**) LG.

**Figure 4 ijms-25-08318-f004:**
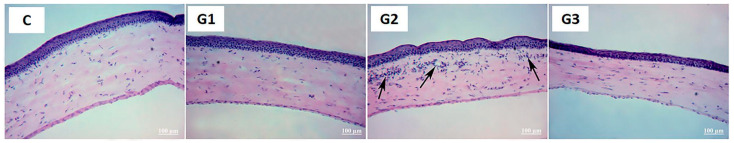
Representative histological image of the cornea of controls (**C**), LG partial ablation (**G1**), ablation followed by allogeneic LG transplant (**G2**), and total ablation (**G3**) at month 2. Samples were stained with H&E (100× magnification). Black arrows indicate leukocytes and microvessels in the cornea stroma.

**Figure 5 ijms-25-08318-f005:**
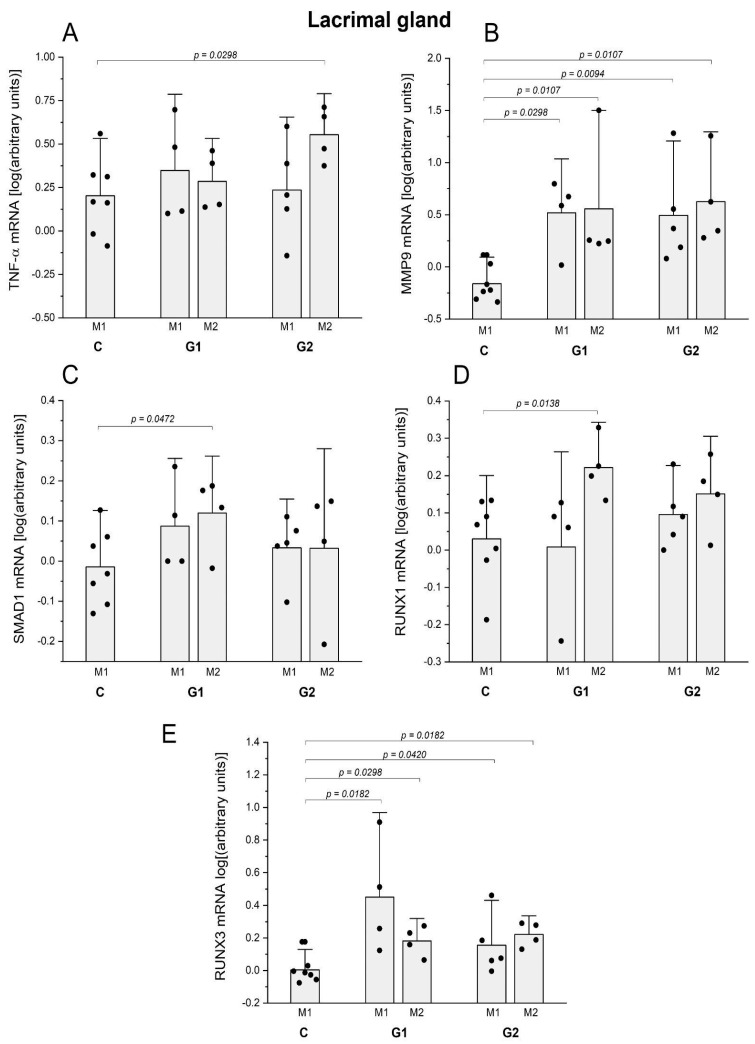
(**A**) Comparative analysis of pro-inflammatory cytokine mRNA of Tnf-α via qPCR in the LG revealed that G2 (partial ablation + allogeneic grafting) presented higher relative mRNA expression of Tnf-α at month 2 (M2). (**B**) Comparative analysis of pro-inflammatory cytokine mRNA of Mmp9 via qPCR in the LG of G1 presented higher relative mRNA expression of Mmp9 at month 1 (M1) (*p* = 0.0298) and at month 2 (M2) (*p* = 0.0107) compared to controls. The G2 (partial ablation + allogeneic grafting) presented higher relative mRNA expression of Mmp9 at M1 (*p* = 0.0094) and at M2 (0.0107). (**C**) Comparative analysis of the mRNA of the pro-mitotic peptide SMAD 1 in the LG showed higher expression in M2 (*p* = 0.0472). (**D**) Comparative analysis of the mRNA of the pro-mitotic peptide of Runx1 in the LG revealed higher levels in G1 at M2 (*p* = 0.0138). (**E**) Comparative analysis of the mRNA of the pro-mitotic peptide Runx3 in the LG of G1 showed higher expression at M1 and M2 (*p* = 0.0182 and 0.0298, respectively) and of G2 at M1 and M2 (*p* = 0.042 and 0.0182). The results are expressed in arbitrary units, normalized to β-actin mRNA). The statistical analysis applied the Mann–Whitney U test, comparing G1, G2, and G3 with the control group (CG). The number of animals was CG 5 rats; G1 M1: 5 rats and G1 M2: 5 rats, G2 M1: 5 rats, and G2 M2: 5 rats. The statistical analysis applied the Mann–Whitney U test, comparing each model versus the control group.

**Figure 6 ijms-25-08318-f006:**
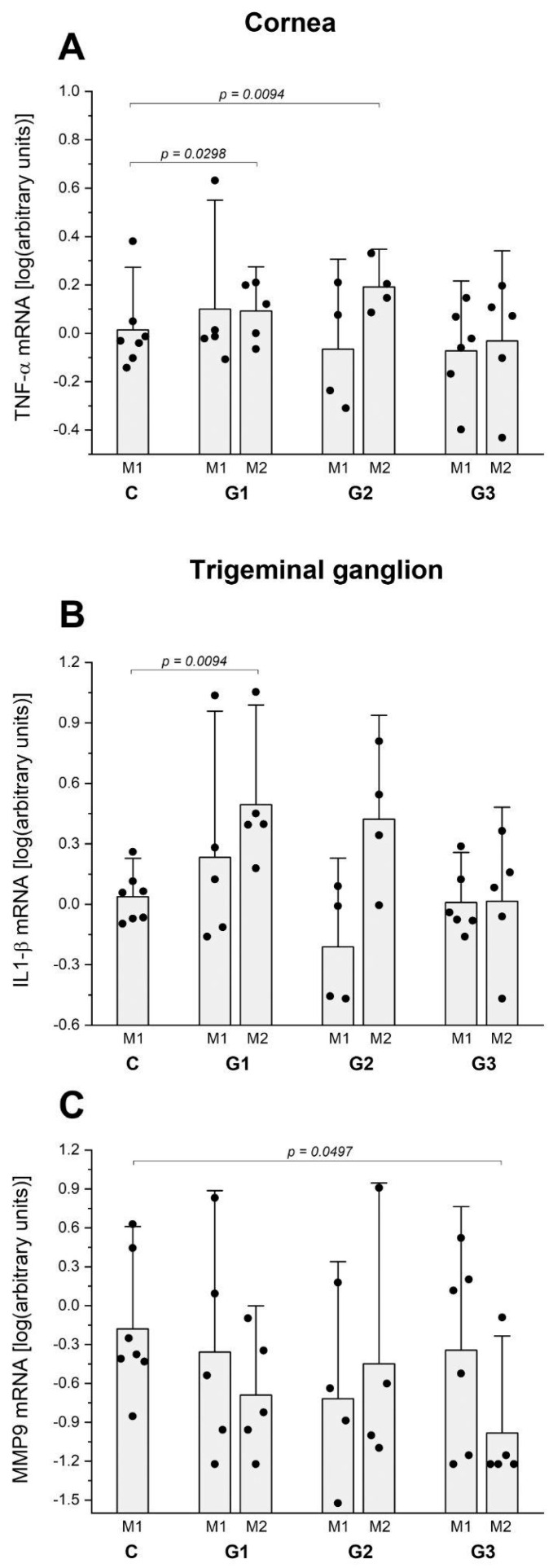
(**A**) Comparative analysis of pro-inflammatory cytokine mRNA Tnf-α via qPCR in the cornea. G1 and G2 presented higher expression of Tnf-α at M2 (*p* = 0.0298 and *p* = 0.094, respectively). (**B**) Comparative analysis of pro-inflammatory cytokine mRNA Il1b via qPCR in the TG. G1 presented higher levels of Il1b at M1 (*p* = 0.0094). (**C**) Comparative analysis of pro-inflammatory cytokine mRNA MMP-9 via qPCR in the TG of G3 revealed lower levels in M2. CG M1: 7 rats; G1 M1: 5 rats and G1 M2: 5 rats, G2 M1: 5 rats and G2 M2: 5 rats; G3 M1: 7 rats and G3 M2: 7 rats. The levels were expressed in arbitrary units, normalized to β-actin mRNA. The statistical analysis applied the Mann–Whitney U test.

**Table 1 ijms-25-08318-t001:** Behavioral and morphometric analysis of body weight, LG weight, and tear flow (phenol red thread test) of the control (C) or the following interventions: LG partial ablation (G1), ablation followed by allogeneic LG transplant (G2), total ablation (G3), n = 4–7 rats/group per time point (mean ± SEM) (Mann–Whitney U test) (#, § indicate *p* < 0.05 versus Control).

	Time	Eye Wipe Test (Mean Wipes in 3 min)	Phenol Red Thread Test (mm)	LG Weight (mg)	*p* versus Control
Control (n = 7 and 5)	1 m	5.6 ± 1.4	8.1 ± 2.0	132.6 ± 15.4	
	2 m		9.2 ± 2.6	125.3 ± 9.2	
G1 (n = 5 per time point)	1 m	2.6 ± 1.2	6.8 ± 1.7	50.3 ± 18.3	
	2 m	3.3 ± 1.3	10.4 ± 2.5	41.3 ± 14.6 #	# 0.0001
G2 (n = 4 per time point)	1 m	14.4 ± 2.2	6.2 ± 1.4	59.7 ± 8.6 #	# 0.0079
	2 m	7.0 ± 1.1	5.8 ± 1.1	45.0 ± 7.8 #	# 0.0001
G3 (n = 6 and 5)	1 m	4.7 ± 2.3	8.2 ± 1.9	N/A	
	2 m	15.3 ± 4.1 §	5.2 ± 1.1	N/A	§ 0.0438

**Table 2 ijms-25-08318-t002:** Antibodies used for Western blot analysis for a comparison of the apoptotic and proliferative response to interventions among LG of experimental groups (G1 to G4) and control rats.

	Catalog Number	Isotype
	Molecular weight (kDa)	Concentration
α-tubulin	Santa Cruz	
SC 8035	Mouse	
Monoclonal	55	200 µg/mL
PCNA	Cell Signaling	
# 2586	Mouse	
Monoclonal	36	200 µg/mL

## Data Availability

The data presented in this study are available on request from the corresponding author.

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
