# Peer review of "Exorbital Lacrimal Gland Ablation and Regrafting Induce Inflammation but Not Regeneration or Dry Eye"

_ijms, 2024, doi:10.3390/ijms25158318_

Round 1

Reviewer 1 Report

Comments and Suggestions for Authors

The study «Exorbital lacrimal gland ablation and regrafting induce local, corneal and trigeminal ganglion inflammation but not regeneration or dry eye» has been submitted to MDPI International Journal of Molecular Sciences.

The article is nicely written and has an accurate, methodological description of each required section and the experiments conducted. Presentation of the results could be further simplified. As molecular pathways are analysed and described, the article is in line with the journal’s scientific focus.

Novelty, significance and rational of the research experiments should be further elucidated in the present study. Firstly, it is not very clear, based on the manuscript, why an animal model for lacrimal gland ablation is necessary from a clinical and scientific point of view. According to reference 1, 23-26, the consequences of a dysfunction or ablation of the lacrimal gland are already known and described in higher-order animals (apes) and humans. Please explain why a further study in rats is required and how it contributes to the understanding of a specific condition?

In addition, it should be explained clearly what disease-condition the ablated glands in this study represent? There are autoimmune diseases (e.g. Sjögren) which lead to a dysfunction of the gland and a destruction process, but is it clear that an ablation properly models such a condition?

To bring this discussion one step further: If the consequences of a destroyed lacrimal gland are known, how does this knowledge contribute towards finding a new treatment? It can be inferred from the following sentence “Still, partial LG ablation and grafting will reveal the proliferative agents and structural attempts to regenerate the LG.” that the idea is to study the molecular mechanism of lacrimal gland ablation and subsequent repair in order to potentially identify a novel target for a therapeutic approach. This is generally a logic way to identify a novel target or treatment mechanism, however, the strategies to identify such a novel target should be better defined and outlined (ideally with a science figure), including a hypothesis specific to this endeavour. This should be the starting point for the research and is also important to justify the animal experiments conducted.

A logic conceptual approach for a study design could be to investigate a specific condition which leads to defect of the lacrimal gland with subsequent dry eye and related complications (provided the lacrimal gland is of primary interest to the researchers). If the condition is insufficiently understood, an animal model could be found for further investigation of a specific potential pathway relevant to the disease mechanism. Subsequently, it could be evaluated if a treatment approach, such as a transplant etc. could be used to improve the condition based on a clear hypothesis from what is understood about the pathology. If this treatment (transplant) is related to complications or not considered effective, an alternative or simplified approach could be derived (for instance stem cell replacement, extracellular vesicles, CRISPR etc.). Please explain the scientific concept on a high level in the introduction for clarification.

Overall, the study has an academic and detailed presentation but should be strengthened in terms of rationale and novelty to be publishable. The animal experiments have to be justifiable to be compliant with international ethics standards.

In text corrections

Title is more a running head, should be shortened and refocused

L264: Please remove “This section may be divided by subheadings….”

P14: there are still comments in the manuscript

Comments on the Quality of English Language

Minor errors but generally solid English.

Author Response

Reply to reviewer 1:

We would like to thank the reviewer for the comment and suggestions on our manuscript “Exorbital lacrimal gland ablation and regrafting induce local, corneal and trigeminal ganglion inflammation but not regeneration or dry eye, renamed as “Exorbital lacrimal gland ablation and regrafting induce inflammation but not regeneration or dry eye”, Manuscript Number: IJMS-3006686. They were taken into consideration, and modifications have been made as described below:

Reviewer comment:

The study «Exorbital lacrimal gland ablation and regrafting induce local, corneal and trigeminal ganglion inflammation but not regeneration or dry eye» has been submitted to MDPI International Journal of Molecular Sciences.

  1. The article is nicely written and has an accurate, methodological description of each required section and the experiments conducted. Presentation of the results could be further simplified. As molecular pathways are analysed and described, the article is in line with the journal’s scientific focus.

Thank you for your comments. We indicate with a lead in the Results session marked in yellow a paragraph to clarify the route of data presentation:

Page 03, line 108: The three distinct models of LG damage are presented compared with naïve controls to allow inference with clinical conditions that simulate partial ablation (G1), partial damage with attempted repair with allogeneic LG transplant (G2), or total LG ablation (G3).

Moreover, we rephrased several sentences and Figure legends of the Results session to improve clarity (yellow marked).

  1. Novelty, significance and rational of the research experiments should be further elucidated in the present study. Firstly, it is not very clear, based on the manuscript, why an animal model for lacrimal gland ablation is necessary from a clinical and scientific point of view.

The present study is required because the animal and in vitro models of lacrimal gland damage and response are still useful to identify the molecular pathways and potential mechanisms involved in promote and inhibit lacrimal gland regeneration. In the adult life, the LG present stem cells and machinery that would allow regeneration, however it stands in a post mitotic and non regenerative status, even after damaged. To clarify this rationale in the present work, the following phrase was added in the Introduction:

Page 02, line 47: LG damage induced by prolonged inflammation in the course of Sjögren’s Disease, cancer, radiotherapy, surgical or traumatic damage to the innervation or the LG itself, among other conditions, induce permanent functional impairment atrophy, fibrosis despite the presence of regenerative conditions [10-12]. The limitations to address why LG persists in the post-mitotic and nonregenerative status, even presenting the machinery for regeneration, justify the present work.  

  1. According to reference 1, 23-26, the consequences of a dysfunction or ablation of the lacrimal gland are already known and described in higher-order animals (apes) and humans.

The reviewer is correct. In humans, LG is a post-mitotic organ. Based on previous works' conclusions, once genetically absent or damaged, the LG is not naturally regenerated or replaced, and the consequences to the ocular surface and the lacrimal functional unit are severe.

  1. Please explain why a further study in rats is required and how it contributes to the understanding of a specific condition?

In adult rodents, the exorbital LG follows the non-regnerative characteristics observed in primates and humans. The studies in rodents are needed because methods to simulate or follow the natural steps of LG diseases and the mechanisms involved in inhibition of regeneration in high-order animals are still very limited. Moreover, the studies in rodents are controversial. As we indicate in the Introduction, lines 40-54, in rodents at early ages or damage induced by inflammatory induction, the LG damage may express some recovery. However, in adult models where the damage is induced by ablation, radiotherapy, nerve or tissue damage by trauma or surgery the regeneration does not occur (Refs 4,5 and 10-15).

  1. In addition, it should be explained clearly what disease-condition the ablated glands in this study represent? There are autoimmune diseases (e.g. Sjögren) which lead to a dysfunction of the gland and a destruction process, but is it clear that an ablation properly models such a condition?

In fact, the LG damage from different causes impairs the function and cause structural changes that at the end are similar, meaning atrophy and fibrosis. The ablated LG, partial and complete intend to simulate the severity of those processes and the possibility to allow auto regeneration in the case of partial ablation. The phrase added in the line 46, page 02, mentioned above address how ablation models these conditions. We hope this insertion clarify the point.

  1. To bring this discussion one step further: If the consequences of a destroyed lacrimal gland are known, how does this knowledge contribute towards finding a new treatment? It can be inferred from the following sentence “Still, partial LG ablation and grafting will reveal the proliferative agents and structural attempts to regenerate the LG.” that the idea is to study the molecular mechanism of lacrimal gland ablation and subsequent repair in order to potentially identify a novel target for a therapeutic approach. This is generally a logic way to identify a novel target or treatment mechanism, however, the strategies to identify such a novel target should be better defined and outlined (ideally with a science figure), including a hypothesis specific to this endeavour. This should be the starting point for the research and is also important to justify the animal experiments conducted.

The present work reveals that LG ablation, with or without grafting induce the expression of pro mitotic elements mRNA, but do not induce cell proliferation and regeneration, even in the long term (2 months). The only technology that demonstrated LG regeneration involved embryonic cells manipulation in vitro and reinsertion. Other alternatives to be investigated may include CRISPR/Cas9 gene edition to reduce the inflammatory and improve the pro mitotic regenerative events after a LG aggression. To improve the discussion, the following phrase was added:

Page 12, line 352: Overall, the present work reveals that LG damage that simulates trauma, surgical intervention, radiotherapy, or long-term disease, as observed in SjD or GvHD that terminate with LG atrophy or fibrosis, is dominated by prolonged inflammatory mediators as TNF-α and MMP-9 and others, and extension of the inflammation to the LFU. Conversely, the expression of pro-mitotic agents in the LG is modest or null, not allowing the regenerative process to progress. The inductive intervention capable of reverting the imbalance between inflammation and regenerative events capable of promoting LG regeneration so far involves in vitro LG organogenesis, using different lineages of allogeneic embryonic stem cells stimulated by humoral factors and further grafting in the location of the damaged LG [19,40]. This strategy to promote pro-mitotic factors and down regulate inflammatory mediators has been proven to be plausible in rodents, although very complex. In the future, strategies involving local cluster regularly interspaced shortly palindromic repeats gene edition using Cas-9 enzyme (CRISPR/Cas9) could support promising therapies to improve the regenerative process and include the expression of Pax6 to allow differentiation in adult LG. However, before considering them for human therapies, CRISPR/Cas9 still needs to be tested in animal models [44].

  1. A logic conceptual approach for a study design could be to investigate a specific condition which leads to defect of the lacrimal gland with subsequent dry eye and related complications (provided the lacrimal gland is of primary interest to the researchers). If the condition is insufficiently understood, an animal model could be found for further investigation of a specific potential pathway relevant to the disease mechanism. Subsequently, it could be evaluated if a treatment approach, such as a transplant etc. could be used to improve the condition based on a clear hypothesis from what is understood about the pathology. If this treatment (transplant) is related to complications or not considered effective, an alternative or simplified approach could be derived (for instance stem cell replacement, extracellular vesicles, CRISPR etc.). Please explain the scientific concept on a high level in the introduction for clarification.

In the manuscript's introduction, we presented the divergences and the need for more information among studies that addressed LG damage and regeneration using different approaches. Then, we indicated that they are justified in proceeding with the present study to understand the LG regenerative process and use this model of LG ablation to evaluate novel alternatives with eye-topic therapies. We hypothesized that if LG ablation induces auto regeneration, this rodent model does not allow its use for DED. Our results demonstrate that DED does not occur, and the LG ablated does not regenerate. In the discussion session, we addressed the potential uses of this novel information and the possible alternatives for LG regeneration, as mentioned in comment # 6. To clarify this point in the Introduction, the following phrase was added to the text:

Page 02, line 89: The study investigates LG ablation. The conflicting information mentioned above reveals that LG ablation or direct damage needs to be more understood regarding DED aspects, natural regeneration and extension of the functional damage in humans and animal models. Therefore, we submitted rats to LG partial ablation, LG ablation with allogeneic LG grafting, or total LG ablation to further investigate the detailed structural and functional repercussions on the LG and the LFU and search for a specific potential pathway relevant to the disease mechanism. This information could be helpful in a subsequent treatment approach, such as an adult LG or organoid transplant, LG gene therapy, or CRISPR to promote genetic edition and pro mitotic agents reprogramming to restore the LG and the LFU homeostasis [19, 30-33].

  1. Overall, the study has an academic and detailed presentation but should be strengthened in terms of rationale and novelty to be publishable. The animal experiments have to be justifiable to be compliant with international ethics standards.

In the comments above, we hope we have clarified the rationale and novelty provided in the present work. The severity of the disease caused by LG damage and severe dry eye has yet to be entirely understood, both in physiopathology and an efficient therapeutic approach. It is associated with blindness and impacts the patient's quality of life, education, labor, and social readaptation. These facts are documented in several clinical studies, as reference #23 in the manuscript and in a recent work from Hung N et al. Revealing risks of corneal damage in aqueous-deficient DED in a 17-year population study in Taiwan (Hung, N et al. Am J Ophthalmol. 2021; 227:231-239) justify the search for scientific investigation. Moreover, the study followed all the animal welfare standards adopted globally. The institutional animal care approved it, and the used committee of the Ribeirao Preto Medical School, University of Sao Paulo, independently revised and approved the study protocol.

The ethic statement was replaced to page 13, line 397 and the rationale to include each of the 04 groups are included at the end of the interventions’ description: Session 4, Methods, page 14: Control (line 412), G1 (line 420), G2 (line 432), G3 (line 437).

  1. In text corrections: Title is more a running head, should be shortened and refocused

We agree and rephrased the Title as follow:

Page 1, line 2: Exorbital lacrimal gland ablation and regrafting induce inflammation but not regeneration or dry eye

  1. L264: Please remove “This section may be divided by subheadings….”

The text was removed.

  1. P14: there are still comments in the manuscript

Thank you. We removed the mentioned comments.

Reviewer 2 Report

Comments and Suggestions for Authors

in their manuscript entitled "Exorbital lacrimal gland ablation and regrafting induce local, corneal and trigeminal ganglion inflammation but not regeneration or dry eye" the autors studied the regenerative responses of the lacrimal functional unit after Lacrimal Gland ablation.

- the experiments fpr this study were well selected.

- presnntation and analysis of the findings are well done, statistical analysis and the presentation in the figures is very well perfomed as well as the figure captions provide all nessesary information.

4. Materials and Methods: the ethics statement line 345  ("All experimental procedures adhered to the ARVO Statement for the Use of Animals in Ophthalmic and Vision Research and were approved by the committee on animal experimentation of the School of Medicine at Ribeirão Preto, University of São Paulo." ) can be stated allready at the beginning of  4.1

The conclusions  (5.)  could be more detailed and should include the limitations of this study (because the study was performed on rats). How can the results beeing transfered on humans ? What is the putlook - future planned experiments?

- 6- Patents : This paragraph can be erased

Comments on the Quality of English Language

manuscript needs a minor work up on English

Author Response

Reply to Reviewer 2:

We would like to express our sincere gratitude to the reviewer #1 for attention and suggestions dedicated to our manuscript IJMS-3006686, renamed as “Exorbital lacrimal gland ablation and regrafting induce inflammation but not regeneration or dry eye”, Manuscript Number: IJMS-3006686. They were taken into consideration, and modifications have been made as described below:

Reviewer comment:

  1. Materials and Methods: the ethics statement line 345 ("All experimental procedures adhered to the ARVO Statement for the Use of Animals in Ophthalmic and Vision Research and were approved by the committee on animal experimentation of the School of Medicine at Ribeirão Preto, University of São Paulo." ) can be stated allready at the beginning of session 4 (Methods).

We attended the recoemdation to replace the phrase in the beggining of the Session 4 (Methods) at line 397, Page 13.

  1. The conclusions (5) could be more detailed and should include the limitations of this study (because the study was performed on rats).

The conclusions were rephrased and the suggestions were included as follow:

Page 13, line 389 (Discussion): In conclusion, exorbital LG ablation did not induce DED or LG repair in rats in this study's ablation models. The possible reasons are the inflammatory disruption induced by the LG damage and the destabilization of the LFU. It may induce local conditions that inhibit the regenerative process. 

Page 17, line 558 (Conclusions): In conclusion, exorbital LG ablation did not induce DED or LG repair in rats in this study's ablation models. The present work has limitations regarding the animal model, age and sex chosen, and follow-up length, and those limitations must be considered in future studies. Nevertheless, the differences in inflammatory responses and tissue injury are damage model type-dependent, not limited to the LG but extended to the TG and cornea. Such variability may have subtle effects on ocular surface health. It should support decisions on future methodological decisions for studies on the topic of the dry eye model and LG regeneration. The future human adult LG regeneration approach should consider multiple approaches, including control of the inflammation and their consequences to the LFU and organogenesis with gene edition.

  1. How can the results beeing transfered on humans ? What is the putlook - future planned experiments?

We addressed this perspective in the Discussion session as also recommended by Reviewer #1 (Line 352, page 12) and in the Conclusion session, as suggested above (line 566, page 17).

  1. Patents : This paragraph can be erased

We erased the item 6 (Patents). Thank you.